# The PI3K/Akt Pathway in Meta-Inflammation

**DOI:** 10.3390/ijms232315330

**Published:** 2022-12-05

**Authors:** Maricedes Acosta-Martinez, Maria Zulema Cabail

**Affiliations:** 1Department of Physiology and Biophysics, Renaissance School of Medicine, Stony Brook University, Stony Brook, NY 11794, USA; 2Biological Science Department, State University of New York-College at Old Westbury, Old Westbury, NY 11568, USA

**Keywords:** meta-inflammation, PI3K, Akt, obesity, macrophages, adipocytes

## Abstract

Obesity is a global epidemic representing a serious public health burden as it is a major risk factor for the development of cardiovascular disease, stroke and all-cause mortality. Chronic low-grade systemic inflammation, also known as meta-inflammation, is thought to underly obesity’s negative health consequences, which include insulin resistance and the development of type 2 diabetes. Meta-inflammation is characterized by the accumulation of immune cells in adipose tissue, a deregulation in the synthesis and release of adipokines and a pronounced increase in the production of proinflammatory factors. In this state, the infiltration of macrophages and their metabolic activation contributes to complex paracrine and autocrine signaling, which sustains a proinflammatory microenvironment. A key signaling pathway mediating the response of macrophages and adipocytes to a microenvironment of excessive nutrients is the phosphoinositide 3-kinase (PI3K)/Akt pathway. This multifaceted network not only transduces metabolic information but also regulates macrophages’ intracellular changes, which are responsible for their phenotypic switch towards a more proinflammatory state. In the present review, we discuss how the crosstalk between macrophages and adipocytes contributes to meta-inflammation and provide an overview on the involvement of the PI3K/Akt signaling pathway, and how its impairment contributes to the development of insulin resistance.

## 1. Introduction

In 1948, the World Health Organization (WHO) recognized obesity as a disease [1], and now defines it as an “abnormal fat accumulation that presents a risk to health”. Since then, obesity has become a global epidemic with a profound health burden and a significant impact on health care expenditures [2,3]. In addition, obesity is associated with the incidence of multiple comorbidities, including cardiovascular diseases and insulin resistance. Insulin resistance contributes to the development of diabetes mellitus type 2 (T2D) and metabolic syndrome, which is a conglomerate of risk factors that includes high blood sugar, excess body fat and abnormal cholesterol [4,5]. A significant underlying cause of obesity-induced insulin resistance is chronic low-grade systemic inflammation [6]. Obesity-induced metabolic inflammation, or meta-inflammation, differs from the classical inflammation paradigm (an acute inflammatory response characterized by cardinal signs such as edema, pain and redness). Instead, metabolic stimuli produce a form of sterile inflammation (non-infectious) that is chronically sustained [7].

Meta-inflammation is a complex multifactorial physiological state characterized by the accumulation of immune cells in the adipose tissue (AT) and the deregulation of AT hormones and cytokines [7,8,9,10]. The AT is a highly insulin-responsive organ that plays an important role in regulating whole-body energy, glucose and lipid homeostasis and endocrine and immune responses. Two of the main cellular players in the AT implicated in the pathogenesis of meta-inflammation are macrophages and adipocytes. Macrophages belong to the innate immune system and comprise the majority of AT-infiltrating cells. Adipocytes are specialized connective tissue cells with roles in energy storage and endocrine functions. The non-resolving inflammatory state in obese AT involves complex intercellular communication between AT-infiltrating macrophages and adipocytes [11]. The obesity-related crosstalk between these two cells intensifies the proinflammatory immune responses in the AT. Cell-to-cell contact, the excessive nutrient environment and paracrine interactions contribute to abnormal secretion of proinflammatory adipokines and cytokines, and the transition of macrophages towards a proinflammatory state [12,13]. 

Immunometabolism is a young field of research that emerged after mounting evidence showed that “obesity impacts the immune system and promotes inflammation” [14]. Because of the known mechanism linking inflammation to obesity-associated complications, there is an increased interest in targeting inflammatory pathways as part of the approach to prevent or control metabolic disorders [15]. One signaling network that is involved in regulating cellular metabolism and inflammatory responses is the phosphoinositide 3-kinase (PI3K)/Akt pathway. 

The PI3K/Akt network modulates glucose metabolism by phosphorylating several metabolic enzymes and regulators of nutrient transport. The PI3K/Akt axis is able to alter inflammatory responses through the recruitment and activation of innate immune cells, such as macrophages [16,17]. In macrophages, the PI3K/Akt pathway transduces signals from various receptors, including insulin receptors (IRs), cytokine and adipokine receptors and receptors essential to induce activation of innate immunity, such as Toll-like receptors (TLRs). Hence, the activation of the PI3K/Akt pathway orchestrates the response to different metabolic and inflammatory signals in macrophages. Furthermore, PI3K/Akt signaling plays an important role in adipocyte physiology, mediating glucose homeostasis and cell differentiation [18]. Thus, impaired PI3K/Akt signaling in the AT is thought to contribute to the non-resolving inflammatory state observed in obesity. 

The aim of this review is to provide an overview of the current knowledge on the mechanisms by which PI3K/Akt signaling regulates obesity-induced inflammation in AT and discuss novel pharmacological strategies to resolve or relapse meta-inflammation. We also explore the paracrine crosstalk between two of the main cellular players implicated in the pathogenesis of meta-inflammation, macrophages and adipocytes, and the role of this interplay in sustaining metabolic inflammation.

## 2. The PI3K/Akt Signaling Pathway, An Overview

PI3Ks were discovered in 1985 [19,20], and have since been implicated in crucial physiological roles, including cell metabolism, survival and cell growth. PI3Ks are lipid kinases classified into three families: Class I isoforms (PI3K α, β, γ, δ), Class II isoforms (PI3KC2 α, β, γ) and Class III isoforms [21,22]. Class I PI3Ks are a family of heterodimeric enzymes consisting of an adaptor and a catalytic subunit that phosphorylate the 3′-OH group of inositol phospholipids. Based on their structure and general mode of regulation, Class I PI3Ks are subclassified into Class IA and Class IB. Class IA PI3K includes p110α, p110β and p110δ, which interact with an SH2-domain-containing p85 regulatory subunit. PI3K negative regulators include the phosphatase and tensin homologue (PTEN), a lipid phosphatase that hydrolyzes phosphatidylinositol 3,4,5-trisphosphate (PIP_3_) to phosphatidylinositol 4,5-biphosphate (PIP_2_), thereby blocking PI3K downstream targets such as Akt. 

PIP_3_ serves as a docking site for signaling proteins that contain a pleckstrin homology (PH) domain, such as Akt (also known as protein kinase B) and PI3K-dependent kinase 1 (Figure 1) [23,24,25]. Akt, the major PI3K downstream effector molecule, is a family of three serine–threonine kinases, the highly homologous isoforms Akt1 (PKB), Akt2 (PKBβ) and Akt3 (PKBγ) [26]. Akt1 is ubiquitously expressed, whereas Akt2 is mainly expressed in insulin-sensitive tissues, such as AT, liver and skeletal muscle, and Akt3 is expressed in the brain and the testes [25]. Akt is composed of three domains: PH, middle kinase and regulatory carboxy-terminal domain. When the PI3K/Akt pathway is activated, such as following the binding of insulin to the IR, the PH domain of Akt associates with PIP_3_, facilitating the phosphorylation of Akt on a threonine in its activation loop by the enzyme 3-phosphoinositide-dependent protein kinase 1 (PDK1). The mammalian target of rapamycin complex 2 (mTORC2) also phosphorylates Akt but on a serine in its C-terminus. The phosphorylation of Akt by these two enzymes grants full activity to Akt. Activated Akt modulates metabolism either directly, by phosphorylating various metabolic enzymes or regulators of nutrient transport, such as glucose transporters (GLUTs), or indirectly, by activating downstream effectors important in cellular metabolic reprogramming, including the mammalian target of rapamycin complex 1 (mTORC1), glycogen synthase kinase 3 (GSK3) and members of the forkhead box O (FOXO) family of transcription factors [27,28]. 

The insulin receptor (IR) is a disulfide-linked homodimer that belongs to the receptor tyrosine kinase (RTK) family of cell surface receptors. Insulin binding to the extracellular region (ectodomain) of IR induces a conformational change that triggers autophosphorylation of several tyrosine residues present in the beta subunit of the cytoplasmic domains, which then prompts the phosphorylation of the insulin receptor substrate (IRS) [29]. Phosphorylated tyrosine residues on IRS act as docking sites for the non-covalent binding of proteins with specific “SH2” domains, such as PI3K and growth-factor-receptor-bound protein 2 (GRB2), which, consequently, activates two main pathways, the PI3K)/Akt pathway and the mitogen-activated protein kinase (MAPK) pathway [30,31]. Akt is responsible for the translocation of the GLUT to the plasma membrane, increasing the glucose influx into the cell and, concomitantly, reducing glycemia.

## 3. The Role of Meta-Inflammation in the Development of Insulin Resistance

Obesity is a major risk factor for T2D, and the fasting hyperglycemia that accompanies T2D is associated with an inadequate insulin response. The Centers for Disease Control and Prevention (CDC) defines a healthy weight range for adult individuals as a body mass index (BMI) between 18.5 and less than 25. After consumption of glucose-rich meals, the pancreatic β cells of these individuals will produce insulin, an endocrine peptide hormone, in response to the increase in circulating glucose levels. Insulin will bind plasma membrane-bound IRs in target cells, provoking a cascade of reactions that will eventually promote the GLUT’s translocation and the subsequent increase in glucose influx into the cell (Figure 1). 

Obesity, on the other hand, is defined by the CDC as adult individuals with a BMI equal to or higher than 30, and overweight as individuals with a BMI range between 25 and less than 30. It is well recognized that overweight and obesity are the most important risk factors contributing to the development of T2D [32]. T2D is caused by the establishment of insulin resistance, where at normal plasma insulin levels, target tissues are unable to mount an integrated glucose-lowering response to insulin. Adipocyte insulin resistance and inflammation have been identified as important contributors to the development of T2D. Insulin resistance can be triggered by various factors, and obesity is one of the most important [32]. Increased basal lipolysis in AT causes elevated levels of free fatty acid in obese patients. The increased concentration of free fatty acids might be responsible for the local production and release of proinflammatory cytokines and chemokines, which promote the infiltration of macrophages into the AT and their concomitant activation [33,34,35]. As obesity progress, there is a dramatic increase in infiltrating macrophages into the AT, in part due to an increase in monocyte chemoattractant protein-1 (MCP-1) expression, one of the most important chemokines regulating migration and infiltration of monocytes/macrophages. The macrophage infiltration leads to a shift from 3% to approximately 20% of total nonadipocyte cells [36]. When monocytes escape the blood stream and enter the obesity-induced metabolically challenged AT, they differentiate to proinflammatory macrophages. The paracrine interaction between activated macrophages and adipocytes further increases the secretion of proinflammatory cytokines and establishes a vicious loop in the obese AT [13]. This paracrine communication promotes the persistent recruitment of macrophages and the expression of proinflammatory cytokine receptors, which further aggravates the vicious spiral [37].The shift in the cellular ratio contributes broadly to the development of the obese AT inflammatory phenotype [38] and, concomitantly, systemic insulin resistance [39]. 

Contributing to the vicious cycle is the formation of crown-like structures (CLSs). CLSs form when AT macrophages (ATMs) accumulate around dead or dying adipocytes. The number of dying adipocytes increases dramatically in obesity. Lindhorst et al. [40] have shown that the threshold size for classic, fast efferocytosis clearance of apoptotic cells and debris from the normal turnover of cells by ATMs is around 25 μm, whereas hypertrophic adipocytes from high-fat-diet (HFD)-fed mice are 5-fold this size, suggesting that classic phagocytosis is not possible, and that the cell remnant after adipocyte apoptosis and secondary necrosis may persist for longer. These findings lead to the hypotheses that CLS formation is a sign of ineffective efferocytosis of dead adipocytes and that this might be another contributing cause of chronic low-grade inflammation in obesity [40].

## 4. The Cellular Players of Meta-Inflammation: Adipocytes and Macrophages

### 4.1. Lean Versus Obese Adipose Tissue

Based on its morphology, location, physiological and functional characteristics, AT can be classified into white (WAT), brown (BAT) or beige (BeAT) subsets. BAT specializes in the production of heat by nonshivering thermogenesis and is usually activated in response to cold exposure [41,42]. Even though the prevalence of BAT in the general population is only around 2.7%, BAT has the capacity to counteract metabolic disease, including obesity and type 2 diabetes. For example, adults with active BAT might have favorable metabolic profiles [43]. On the other hand, WAT stores excess energy as triglycerides, which are broken down via lipolysis to supply energy during fasting periods. WAT is the most abundant AT and includes subcutaneous adipose tissue (SAT) and visceral adipose tissue (VAT). WAT is also considered an endocrine organ, secreting adipokines such as leptin, adiponectin, resistin, C-C Motif Chemokine Ligand 2 (CCL2) and IL-6. Nutritional imbalance influences WAT metabolism and immune functions and, as a consequence, the circulating levels of adipokines. For example, leptin alters central food intake [44] and increases energy expenditure [45,46], and its production by the AT facilitates the secretion of proinflammatory cytokines [37,47]. On the other hand, adiponectin regulates glucose levels, lipid metabolism and insulin sensitivity through its anti-inflammatory effects [48]. An imbalance between energy intake and energy expenditure results in weight gain and obesity. If the caloric excess is sustained, the need to increase lipid storage will lead to adipocytes’ expansion in cell size (hypertrophy) and number (hyperplasia) [49]. 

In metabolically unhealthy obese individuals, the storage capacity of SAT, the largest WAT depot, is limited, and further caloric surplus drives the fat accumulation in ectopic tissues (e.g., liver, skeletal muscle and heart) and in the visceral adipose depots, promoting lipotoxicity. Excessive ectopic lipid accumulation leads to local inflammation and insulin resistance. The uncontrolled inflammatory responses in WAT contribute to the development of chronic low-grade inflammation, and therefore promote the evolution of systemic insulin resistance [50]. It has been proposed that adipocyte hypertrophy and its associated cell stress and cell death initiates visceral AT inflammation [51], which is linked to insulin resistance and ultimately the development of T2D [8,52].

#### Inflammation-Mediated Impairment of PI3K/Akt Signaling in Adipocytes

In adipocytes, glucose uptake is regulated by insulin–insulin receptor signaling, which controls the subcellular location of GLUT4, the main glucose transporter present in these cells [53]. During low-insulin states, GLUT4 is intracellularly sequestered within fat cells, and is translocated to the cell surface upon insulin stimulation [54,55]. In AT, glucose uptake is regulated by PI3K/Akt signaling pathway mediating downstream insulin signaling, promoting lipid biosynthesis and suppressing lipolysis by inhibiting protein kinase A (PKA), and by inhibiting FOXO1-induced lipolysis. For example, the adipose-cell-specific deletion of the Class 1A PI3K catalytic subunit p110α showed that AT-specific loss of p110α resulted in a decrease in insulin-stimulated PI3K activity associated with IRS-1. Moreover, insulin-induced phosphorylation of Akt Thr^308^ (a site controlled by PI3K via PDK1) was also attenuated in the α−/− adipocytes. Correspondingly, inhibition of PI3K/Akt signaling decreases insulin sensitivity by degrading Sort1, an element of the GLUT4 storage vesicles [56,57]. Therefore, these studies suggest that p110α plays an important role in insulin-stimulated PI3K/Akt signaling in WAT, and that Class 1A PI3K catalytic subunits play a critical role in the regulation of energy expenditure [42]. 

A proinflammatory microstate involving the presence of several proinflammatory cytokines, including tumor necrosis factor alpha (TNF-α), interleukin-6 (IL-6) and interferon gamma (INF-γ), stimulate the expression of a suppressor of cytokine signaling (SOCS)3 in adipocytes [58,59]. SOCS3 belongs to the SOCS family of proteins, which are characterized by their ability to cause feedback inhibition of cytokine and growth factor signaling by negatively regulating the action of the cytokine-induced Janus kinase (JAK)/signal transducer and activator of transcription (STAT) pathway [60]. Insulin resistance can be induced by the overexpression of SOCS3 in the AT of obese mice [58]. In addition, SOCS3 inhibits insulin signaling by targeting IRS-1 and IRS-2 for ubiquitin-mediated degradation [61]. Overexpressing SOCS3 in AT (aP2- SOCS3 mouse) was found to decrease IRS-1 protein levels with a subsequent decrease in insulin-stimulated IRS-1 and -2 phosphorylation, leading to a decrease in p85 binding to IRS-1, which subsequently resulted in lower PI3K/Akt activation and a subsequent decrease in insulin-stimulated glucose uptake in adipocytes [62]. Conversely, a SOCS3-deficient adipocyte model found that the activity of IRS-associated PI3K was enhanced, causing an increased insulin-stimulated glucose uptake [59]. 

Recent findings suggest that angiotensin II (Ang II) attenuates insulin signaling in AT through enhancement of SOCS3 [59]. Ang II is the main bioactive peptide of the renin–angiotensin system (RAS) [63] and it is thought to contribute to insulin resistance by promoting the proteasome-mediated degradation of IRS-1 [64,65]. In addition, Ang II impairs insulin-induced Akt activation and its downstream substrates, AS160 and GSK-3β, in both isolated rat adipocytes [66] and differentiated adipocytes from the cell line 3T3-L1 [67]. These findings show that SOCS3 and Ang II indirectly impaired PI3K/Akt-induced glucose uptake in adipocytes by targeting the PI3K/Akt upstream modulators IRS-1 and IRS-2 for degradation by the ubiquitin system.

### 4.2. Macrophage Phenotypic Dynamics in Response to the Microenvironment

Macrophages constitute the first line of defense for pathogenic insults or tissue damage initiating the inflammatory cascade. In the classical paradigm, macrophages adopt two distinct activation phenotypes, the proinflammatory and classically activated (M1) phenotype, and the anti-inflammatory and alternatively activated (M2) phenotype [68]. These two activation spectra display distinct surface markers, secrete different cytokines and are associated with distinct gene expression profiles, providing their unique phenotypes [69,70,71]. M1 macrophages induce the acute phase of the inflammatory response by producing proinflammatory cytokines, including TNF-α, IL-6 and interleukin-1β (IL-1β). M1 macrophages also exhibit an increased expression of inducible nitric oxide synthase (iNOS), inflammasome activation and enhanced endoplasmic reticulum (ER) stress [72,73,74]. The inflammasomes are innate immune system sensors that induce inflammation and regulate the activation of caspase-1, a proteolytic enzyme that cleaves proinflammatory cytokines into their active forms [75,76]. On the other hand, M2 macrophages are characterized by the expression of peroxisome-proliferator-activated receptor gamma (PPARγ), which promotes tissue remodeling and aids in resolving inflammation [77]. The M2 phenotype also activates the expression of Krüppel-like factor 4 (KLF4), an evolutionarily conserved zinc-finger-containing transcription factor that regulates various cellular processes. KLF4 regulates macrophage polarization towards an anti-inflammatory state by mediating the gene transcription of M2 macrophage gene signatures, which include arginase 1 or resistin-like α or are found in inflammatory zone protein (Retnla/Fizz1) and chitinase-like protein 3 (Chi3l3) [78]. Arginase 1 is one of the hallmarks of alternatively activated macrophages and contributes to the resolution of inflammation and tissue repair by degrading arginine and preventing the generation of cytotoxic nitric oxide [79]. Retnla/Fizz1 and Chil3/Ym1 are also induced during macrophage M2 activation [80]. However, it is important to point out that these are not all-or-none phenotypes; rather, these constitute an M1/M2 spectrum modulated by the tissue of residence and the specific microenvironment.

Macrophages are also influenced by the local microenvironment. For instance, M1 macrophages generated in vitro do not express the membrane marker CD11c, which is a marker of M1 macrophage activation; however, if bone-marrow-derived macrophages (BMDMs) are differentiated in the presence of adipocytes, then CD11c is induced [33,73]. Resident ATMs have a distinctive phenotype compared to other tissue macrophages [81]. In a lean AT, resident macrophages adopt an anti-inflammatory M2 phenotype expressing high levels of CD206, also known as mannose receptor C type 1 (MRC1) [82]. In contrast, in an obese state, ATMs are activated by diverse metabolic stimuli including free fatty acids, high insulin and high glucose levels, and they display surface markers that are unique to this state. These metabolically activated macrophages (MMes) do not completely resemble a classical M1 activation nor an alternative M2 activation; instead, in response to being metabolically challenged, MMes exhibit distinct markers [69,77]. For example, ATMs from obese mice express low levels of CD206 and high levels of integrin CD11c [72,73,83]. A recent study evaluating VAT from bariatric surgery patients found that CD64+ ATMs, a marker that distinguishes macrophages from dendritic cells, were also CD206+ and secreted both proinflammatory and anti-inflammatory cytokines [84]. Moreover, the number of CD64+–CD206+ ATMs was increased in obese individuals compared with normal-weight individuals [84]. 

Macrophage activation in the context of obesity-associated inflammation is also governed by epigenetic mechanisms, where chromatin accessibility to transcription factors determines repression or suppression of inflammatory genes. For instance, the G-protein pathway suppressor 2 (GPS2) is part of a co-repressor complex that inhibits inflammatory gene activation [85], and its expression is also representative of the M2 macrophage phenotype. GPS2-deficient mice fed an HFD presented a state of increased systemic inflammation and impaired glucose tolerance, suggesting that GPS2 levels negatively correlate with systemic and AT inflammation [86]. On the other hand, transplantation of GPS2-overexpressing bone marrow into ob/ob and HFD-induced obesity mouse models reduced inflammation and improved insulin sensitivity [46,86]. 

The establishment of the MMe phenotype involves the activation of several signaling pathways. Under a caloric excess environment, saturated fatty acids (SFAs) mediate macrophage inflammation by activating TLR4 [77,87]. Activation of TLR4 stimulates the proinflammatory Jun N-terminal kinase (JNK) and inhibitory-κB kinase (IKK) pathways [7]. The PI3K/Akt axis is also activated by TLR4, as well as other pathogen recognition receptors, cytokines and chemokines and Fc receptors, thus regulating downstream signals that control the production of cytokines [88]. The activation of PI3K/Akt signaling is crucial to restrict inflammation and to promote anti-inflammatory responses in TLR-induced macrophages and has long been considered as a negative regulator of the TLR and Nuclear factor kappa-light-chain-enhancer of activated B cell (NF-κB) signaling in macrophages [88,89], contributing to macrophage polarization (Figure 2). NF-κB is a master inflammatory transcriptional factor involved in the expression of several proinflammatory genes and a key regulator of the ATM M1 program. 

In summary, the ATM phenotypic dynamics can vary considerably depending on different contexts, including the levels of macronutrients such as free fatty acids, insulin and glucose, the cellular interactions, the activation of specific proinflammatory signaling such as JNK/IKK pathways and the tissue of residence.

#### 4.2.1. PI3K/Akt in Metabolic-Activated Macrophages

The obesity-induced metabolic inflammatory state triggers the activation of macrophage pattern recognition receptors (PRRs)—including the immune and nutrient status sensor TLRs—inflammasomes and the nucleotide oligomerization domain (NOD) [76]. Elevated levels of fatty acids are caused by increased basal lipolysis in the obese AT. High concentrations of SFAs stimulate TLR4 and as a consequence the proinflammatory pathways JNK and IKK. Both JNK and IKK phosphorylate the serine/threonine residues on IRS proteins (Figure 2). This phosphorylation prevents the IR from interacting with the IRS, blocking the downstream insulin signal transduction, including the PI3K/Akt pathway. Therefore, phosphorylation of the IRS on inhibitory sites by JNK and IKK promotes insulin resistance, inhibits glucose uptake by the cell and as a consequence leads to hyperglycemia [83,90]. 

In obesity, the activity of JNK and IKKβ is elevated. These serine kinases participate in the production of inflammatory cytokines via the transcription factors AP-1 and NF-κB, respectively. The produced proinflammatory cytokines (e.g., TNF-α, IL-6, IL-1β) are able to activate JNK and IKKβ in an auto-paracrine manner, leading to an inflammatory amplification loop [91]. Proinflammatory cytokines also activate other kinases, such as mTOR/S6 kinases and MAP kinases, as well as SOCS proteins, which also interfere with insulin signaling and adipocyte function [90]. The increase in circulating insulin, amino acids and proinflammatory cytokines observed in chronic overnutrition results in mTORC1 activation, which mediates feedback inhibition of the PI3K/AKT pathway through the phosphorylation of the inhibitory serine/threonine residues on the IRS [92].

#### 4.2.2. The Metabolic Reprogramming of Macrophage Polarization

Macrophage metabolic reprogramming refers to the ability of macrophages to alter their metabolism in order to support their functional activation state (also known as polarization). In general, proinflammatory macrophages shift to aerobic glycolysis to meet cellular energy demands and support the cell during acute inflammation. In contrast, anti-inflammatory macrophages shift towards fatty acid oxidation, generating high amounts of ATP via oxidative phosphorylation [93,94]. However, several studies suggest that macrophage metabolic reprogramming is much more complex and depends in part on the activating stimulus [94].

Upon activation, proinflammatory macrophages experience a “respiratory/glycolytic burst” to support the synthesis of metabolites and increase the phagocytic capacity that follows macrophage stimulation [6]. Although macrophages express glucose transporters GLUT1, 3 and 5, GLUT1 is the predominant isoform controlling glucose uptake. Macrophage metabolic reprograming is driven by GLUT-1-mediated glucose uptake and an elevated rate of glycolysis [95]. The activation of macrophages to an M1 program is a metabolically expensive event and, as a consequence, glucose uptake is increased. For example, lipopolysaccharides (LPSs) induce the activation of TLR4, which triggers the biosynthesis of the proteins, lipids and nucleic acids required by macrophages to respond to infection or tissue injury during the inflammatory response. Thus, proinflammatory macrophages exhibit a metabolic shift to an elevated rate of glycolysis when activated through TLRs. This leads to the production of various proinflammatory cytokines, such as IL-1β and TNF-α, and involves an increase in GLUT1 expression [96]. GLUT1 expression, and consequently the increase in glucose uptake, is due to the activation of the transcription factor NF-κB (Figure 2). PI3K/Akt activation is involved in the induction of GLUT1 expression by stabilizing the transcription factor Hypoxia-inducible factor 1-alpha (HIF-1α), which has been found to be elevated in LPS-treated macrophages [97]. HIF-1α contributes to glycolysis by upregulating the expression of glucose transporters and glycolytic enzymes during inflammation [98]. In addition, HIF-1α induces the expression of proinflammatory genes, notably *Il1b*, which encodes cytokine IL-1β [99]. In macrophages exposed to hypoxic stress, activation of the PI3K/Akt pathway results in the upregulation of TLR4 expression via activation of HIF-1α [100].

The PI3K/Akt signaling pathway also regulates macrophage polarization though mTOR [101,102]. mTOR is a serine/threonine kinase that acts as a cellular nutrient and energy sensor. mTOR signals via two complexes, mTORC1 and mTORC2, each responding to different stimuli; mTORC1 is rapamycin-sensitive, being regulated by PI3K and MAPK pathways through the activation of receptor tyrosine kinases, including the IR (Figure 2) [103]. Polarization of M1 macrophages through LPS-TLR4 stimulation triggers the proinflammatory IKK/NF-κB cellular pathway, inhibiting the tuberous sclerosis protein complex (TSC) and the negative regulator of mTOR and, consequently, increasing activation of mTORC1. For example, inhibition of the mTORC1 complex results in enhanced M1 macrophage proinflammatory responses both in vitro (using rapamycin) and in vivo (using mice with myeloid-lineage-specific Raptor deletion to disrupt mTORC1 functions) [103]. The TSC1/2 complex is a critical upstream negative regulator of mTOR and is also involved in the regulation of macrophage polarization [104]. For instance, using a mouse model with a myeloid-lineage-specific deletion of TSC1, Byles et al. [101] showed that Tsc1(Δ/Δ) macrophages have constitutive activation of mTORC1, downregulated Akt activity and enhanced proinflammatory and reduced anti-inflammatory cytokine production. It was suggested that the reduced Akt activity after TSC1 deletion in macrophages may be due to mTORC1-mediated negative feedback of Akt signaling (Figure 1). mTORC1 is highly active in metabolically demanding situations such as LPS-TLR4 stimulation, and its activity is upregulated during nutrient excess environments, such as the one observed in obese AT [105].

## 5. Macrophage–Adipocyte Crosstalk: Contribution of Impaired Autophagy to Adipose Tissue Insulin Resistance

Hallmarks of AT disfunction in obesity include adipocyte hypertrophy, immune cell infiltration, especially macrophages, and increased expression of proinflammatory cytokines and adipokines [106]. Several studies have investigated the paracrine and cell-to-cell interaction between adipocytes and ATMs in a metabolic inflammatory state to elucidate the impact of this interplay in the pathogenesis of the non-resolving meta-inflammation. For instance, the serum-retinol-binding protein 4 (RBP4) might play a role in either initiating or sustaining the proinflammatory state of obese AT. RBP4 is a proinflammatory adipokine, and its circulating levels are increased in insulin-resistant states, during AT expansion and in obese humans and mouse models of obesity [107]. RBP4 induces the expression and secretion of proinflammatory cytokines in primary human macrophages as well as in mouse macrophages by activating the proinflammatory NF-κB and JNK signaling pathways [108]. Using cocultured 3T3L1 adipocytes and RAW264.7 macrophages in a noncontact system, it was found that RBP4 inhibits insulin-dependent Akt phosphorylation in adipocytes by inducing proinflammatory cytokines in macrophages. Accordingly, a study using a coculture contact model of differentiated 3T3 adipocytes (dif3T3s) with RAW264.7 macrophages found that the phosphorylation of Akt in dif3T3 was decreased, suggesting that inflammation-activated macrophages reduce insulin action in adipocytes [109]. These investigations suggest that in a nutrient-rich AT microenvironment, secretion of proinflammatory adipokines such as RBP4 promotes the production of proinflammatory cytokines by ATMs. The interaction between proinflammatory cytokines and their cognate cytokine receptors in adipocytes might lead to a hyperactive mTORC1 pathway, which ultimately disrupts the insulin-induced IRS-PI3K/Akt axis, preventing glucose uptake and therefore promoting insulin resistance [37].

The obesity-induced proinflammatory milieu in AT impairs macrophage autophagy [110]. Autophagy, also known as macroautophagy, is an intracellular degradation process that maintains cellular homeostasis through degradation of abnormal proteins and damaged organelles. Autophagy begins with the inhibition of mTOR or the activation of 5′-AMP-activated protein kinase (AMPK). The activation of AMPK and mTORC1 is also coupled with the translocation of transcription factor RB (TFEB), which augments autophagy and lysosomal biogenesis [111]. Membrane nucleation and phagophore formation are followed by elongation and maturation before autophagosome fusion with the lysosome for cargo degradation and recycling (for a detailed review of the autophagy cellular process, see [112]). AMPK and mTORC1 are themselves activated by cellular and metabolic stress. AMPK is activated when cellular energy (ATP) levels are low, while nutrient abundance activates mTORC1. Unc-51-like kinase 1 (ULK1) and 2 (ULK2) are needed for autophagy initiation [113]. AMPK and mTORC1 exert opposing effects on ULK1; while ULK1 is activated by AMPK-mediated phosphorylation, it is inhibited by mTORC1-mediated phosphorylation [113]. Therefore, autophagy is activated by starvation while it is suppressed when nutrient availability is high. 

Autophagy has a key role in adipogenesis as ablation of key autophagy genes reduces weight gain in obese mouse models [114,115]. Obesity interferes with the autophagy process; excess nutrition can suppress initiation of autophagy, and lipotoxic insults such as inflammation interfere with autophagy by inhibiting autophagosome degradation [116]. Genetic studies in which the expression of autophagy-related genes was found to be altered support the negative impact of obesity on autophagy. For instance, systemic (global) overexpression of Atg5, which upregulates autophagy, protected mice from age-associated obesity and insulin resistance [117]. On the other hand, pharmacological activation of autophagy, through rapamycin or carbamazepine, reduces steatosis and improves insulin sensitivity in HFD-induced non-alcoholic fatty liver disease [118].

Autophagy is required to suppress proinflammatory M1 and promote anti-inflammatory M2 polarization in macrophages [110], and has an anti-inflammatory action exerted through the downregulation of the inflammasome [119,120] (Figure 3). The anti-inflammatory effects of autophagy are also supported by studies showing that HFD-induced obesity is associated with decreased macrophage autophagy [114]. ATG7 is an E1-like ligase that plays a central role in autophagosome biogenesis [121]. Studies using Atg7KO mice showed that, in obesity-induced inflammatory conditions, autophagy is impaired in ATMs, increasing the production of reactive oxygen species (ROS), which provokes the release of proinflammatory cytokines, further aggravating inflammation and autophagy impairment. The inflammasome-mediated increases in ROS in turn impair insulin signaling. Concomitantly, the increase in inflammatory cytokines inhibits adipocyte Akt. The inhibition of adipocyte Akt signaling is likely induced by downregulated mTORC1 activity after proinflammatory cytokine–cytokine receptor interactions. Activation of mTORC1 is known to promote the phosphorylation of IRS on inhibitory sites, thereby interrupting the activation of PI3K/Akt signaling and therefore promoting adipocyte insulin resistance [122] (Figure 3).

Even though a defective autophagy can result in impaired metabolic homeostasis, its inhibition has also been shown to alter the detrimental sequelae of obesity. For example, targeted deletion of *Atg7* in AT resulted in lean mice with decreased white adipose mass and enhanced insulin sensitivity; it also resulted in browning of WAT and an increase in normal BAT, leading to an elevated rate of fatty acid, β-oxidation and a lean body mass [114]. Additionally, knockdown of Atg7 or Atg5 in 3T3-L1 preadipocytes resulted in the inhibition of lipid accumulation and decreased protein levels of adipocyte differentiation factors [57], indicating that autophagy also has a key role in adipogenesis. In other tissues, inhibition of autophagy upregulates compensatory pathways that are protective against obesity and obesity’s detrimental effects; for instance, mice with skeletal-muscle-specific deletion of *Atg7* showed decreased fat mass and were protected from diet-induced obesity and insulin resistance [123]. This phenotype was accompanied by increased fatty acid oxidation and browning of WAT owing to induction of fibroblast growth factor 21 (Fgf21). Hence, as a result of autophagy deficiency and subsequent mitochondria dysfunction, the compensatory induction of Fgf21 expression promotes protection from diet-induced obesity and insulin resistance. These studies suggest that whether a protective or pathologic phenotype is observed as a result of autophagy deficiency will highly depend on the cell type involved.

## 6. Conclusions

The mechanism of immunometabolic dynamics is very complex and involves the participation of multiple cellular signaling and metabolic networks as well as paracrine and autocrine interactions. In this review, we focused on the PI3K/Akt signaling pathway, a major intracellular network that plays an important role in metabolic regulation and immune system homeostasis, and its impact on the non-resolving inflammation induced by obesity. By promoting glucose utilization, protein synthesis and lipogenesis, the PI3K/Akt pathway has a key role in the maintenance of adipocyte metabolism. However, in the obese AT environment, increased levels of proinflammatory cytokines promote an interruption of IRS-PI3K/Akt signaling, preventing the translocation of GLUTs and thus decreasing glucose uptake by adipocytes and promoting insulin resistance (Figure 4). A major contributor to this impairment is the metabolic-induced proinflammatory ATM phenotype (MMe); the excess of SFAs activates TLR4-NF-κB and inhibits TSC1/2, prompting a hyperactive mTORC1 and, consequently, impairing ATM autophagy. Without the regulatory action of autophagy, the inflammasome is rendered active, promoting the MME phenotype and the obesity-induced non-resolving inflammation (Figure 4).

Clinical strategies that can resolve meta-inflammation are essential, as these therapeutics might provide immense clinical, social and economic benefits. The JAK/STAT signaling pathway is an emergent player in metabolism and has been shown to be dysregulated in obesity and T2D [124]. Recently, baricitinib, a selective JAK1/2 inhibitor used to treat rheumatoid arthritis [125], has been investigated for the treatment of T2D and its complications. Barcitinib decreased inflammatory biomarkers and restored insulin signaling in the liver and skeletal muscle of HFD-fed mice by targeting JAK/STAT-mediated PI3K/AKT signaling [126,127]. This small molecule inhibitor blocks cytokine signaling by suppressing the phosphorylation of the transcription factor STAT-1. Inhibition of STAT-1 might decrease the transcription of SOCS, thus preventing the proteasomal degradation of IRS1, and restoring IRS1-PI3K/AKT signaling in T2D. Repurposing JAK inhibitors already approved to treat chronic inflammatory diseases has the potential to decrease the detrimental action of inflammatory cytokines in obesity-induced chronic inflammation. Therefore, pharmacological modulation of the JAK/STAT cascade to treat meta-inflammation warrants further investigation. 

A valuable research perspective is given by the investigation of molecular and cellular interplays between two of the main players in meta-inflammation, macrophages and adipocytes. The examination of the delicate intercellular connection in the context of obesity could offer novel insights that would guide the intelligent design of therapeutic strategies to resolve the persistent inflammation associated with obesity.

## Figures and Tables

**Figure 1 ijms-23-15330-f001:**
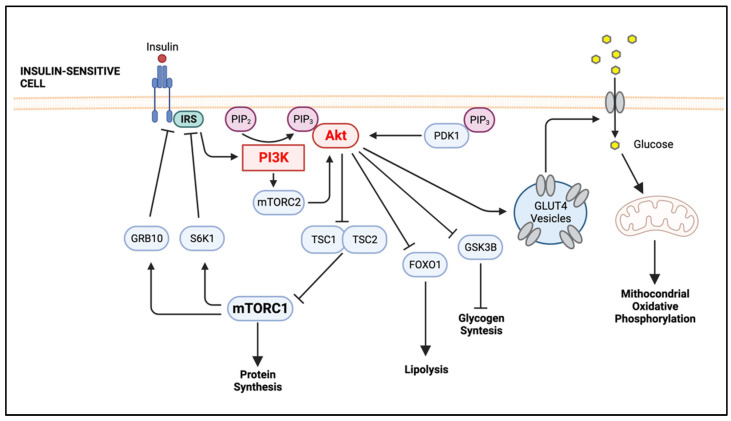
Insulin Signaling in Insulin-responsive Cell. In healthy individuals, insulin binds to the ectodomain of IRs, causing a conformational change that leads to the autophosphorylation of the tyrosine residues in the cytoplasmic β subunits. The IRS then binds to the phosphorylated tyrosine residues of IRs, causing them to be phosphorylated. Phosphorylated IRSs bind to PI3K, and as a consequence, PI3K is activated and converts PIP2 to PIP3. PIP3 recruits the serine/threonine kinase Akt (also known as protein kinase B) to the plasma membrane, where it is activated by phosphorylation. The activated Akt is responsible for the translocation of GLUT4 to the plasma membrane, allowing the entry of glucose into the cell, which will be used to produce ATP through mitochondrial oxidative phosphorylation. Activated Akt also promotes the inhibition of GSK3B, which leads to higher glycogen synthesis. Once activated, Akt also inhibits transcription factor FOXO1, which promotes transcription of lipoprotein lipase, an enzyme responsible for the breakdown of triglycerides. Furthermore, phosphorylated Akt inhibits TSC1/2, and as a consequence, mTORC1 is activated, which leads to protein synthesis. Activation of S6K1 by mTORC1 phosphorylates inhibitory serine residues on the IRS, while mTORC1-induced activation of GRB10 prevents insulin signaling by binding to IR. (Created with BioRender.com on 19 October 2022). Abbreviations: IRs, insulin receptors; IRS, Insulin receptor substrate; PI3K, phosphoinositide 3-kinase; PDK1, 3-phosphoinositide-dependent protein kinase 1; PIP_2_, phosphatidylinositol 4,5-biphosphate; PIP_3_, phosphatidylinositol 3,4,5-trisphosphate; GLUT4, glucose transporter type 4; GSK3B, glycogen synthase kinase 3β; FOXO1, forkhead box O; TSC1/2, tuberous sclerosis protein complex 1 and 2; mTORC1, mammalian target of rapamycin complex 1; mTORC2, mammalian target of rapamycin complex 2; S6K1, S6 kinase 1; GRB10, growth-factor-receptor-bound protein 10.

**Figure 2 ijms-23-15330-f002:**
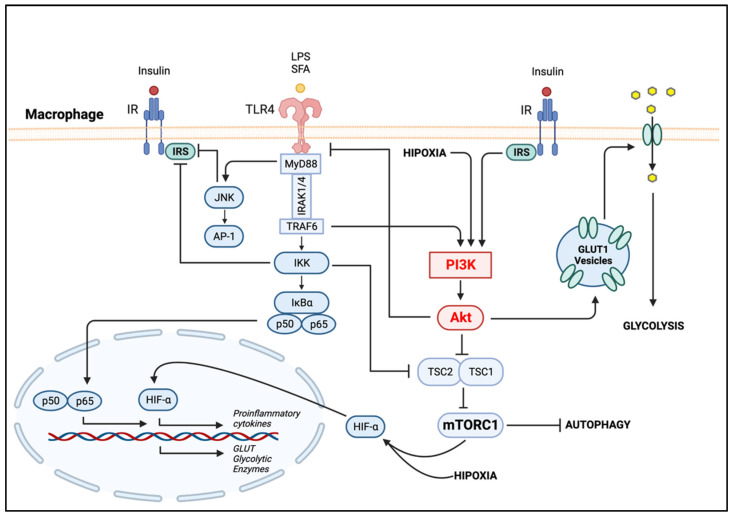
PI3K/Akt Signaling in Adipose Tissue Macrophages. Bacterial LPSs or SFAs stimulate TLR4, which recruits the adapter protein MyD88. Following activation of the TLR4/MyD88 axis, IRAK1 and IRAK4 are recruited and interact with TRAF6 proteins, which subsequently phosphorylate the IKK complex (IKKβ, IKKα and NEMO). The activation of IKK further phosphorylates IκB proteins, leading to its ubiquitination and proteasomal degradation and freeing NF-κB complexes (p50–p65). NF-κB then translocates to the nucleus, where it regulates the expression of proinflammatory genes. Activation of IKK inhibits TSC1/2, the negative regulator of mTORC1, rendering the later active. The upregulation of mTORC1 activity inhibits autophagy. In a state of hypoxia, HIF-1α is stabilized by the activation of PI3K/Akt. HIF-1α upregulates the expression of glycolytic enzymes and glucose transporters contributing to inflammation-induced glycolysis. Activation of TLR4 also stimulates the proinflammatory JNK pathway, which, along with IKK, phosphorylates the IRS on inhibitory sites. (Created with BioRender.com on 19 October 2022). Abbreviations: LPS, lipopolysaccharide; SFAs, saturated fatty acids; TLR4, Toll-like receptor 4; MyD88, myeloid differentiation factor 88; IRAK1/4, IL-1 receptor-associated kinase 1 and 4; TRAF6, tumor necrosis factor receptor associated factor 6; PI3K, phosphoinositide 3-kinase; IKK, IκB kinase; IκB, inhibitor κB kinase; NF-κB, Nuclear factor kappa-light-chain-enhancer of activated B cells; TSC1/2, tuberous sclerosis protein complex 1 and 2; mTORC1, mammalian target of rapamycin complex; HIF-α, hypoxia-inducible factor 1-α; GLUT1, glucose transporter type 1; IR, insulin receptor; IRS, Insulin receptor substrate, JNK, Jun N-terminal kinase; AP-1, activator protein-1.

**Figure 3 ijms-23-15330-f003:**
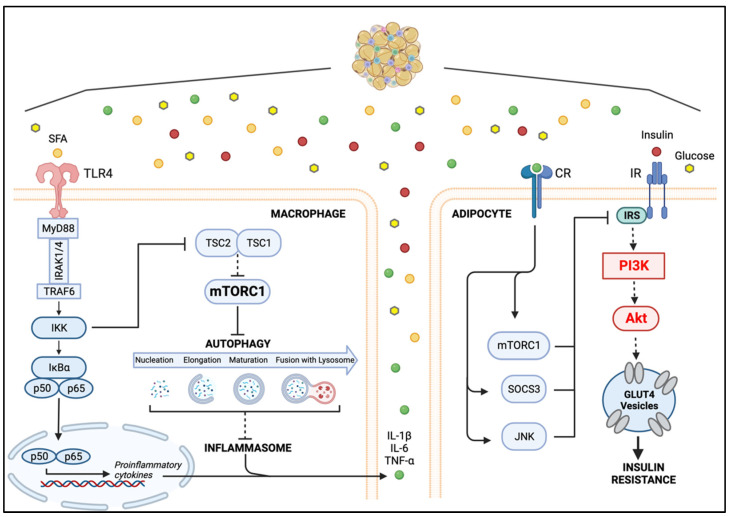
The Macrophage–Adipocyte Crosstalk in Metabolically challenged Adipose Tissue. Activation of macrophage TLR4 by SFAs induces the expression of proinflammatory cytokines, such as IL-1β, IL-6 and TNF-α. Binding of proinflammatory cytokines to their corresponding receptors on adipocytes induces the expression of SOCS3 and activates JNK and mTORC1. Stimulation of SOCS3 leads to proteasomal degradation of IRS1. Activated JNK and mTORC1 block IRS1 downstream signaling by promoting phosphorylation of inhibitory serine residues on IRS1. The concomitant insulin signaling inhibition promotes the establishment of insulin resistance in the AT. (Created with BioRender.com on 19 October 2022). Abbreviations: SFAs, saturated fatty acids; TLR4, Toll-like receptor 4; MyD88, myeloid differentiation factor 88; IRAK1/4, IL-1 receptor-associated kinase 1 and 4; TRAF6, tumor necrosis factor receptor associated factor 6; IKK, IκB kinase; IκB, inhibitor κB kinase; NF-κB, Nuclear factor kappa-light-chain-enhancer of activated B cells; TSC1/2, tuberous sclerosis protein complex 1 and 2; mTORC1, mammalian target of rapamycin complex; GLUT4, glucose transporter type 4; IR, insulin receptor; IRS, Insulin receptor substrate; PI3K, phosphoinositide 3-kinase; JNK, Jun N-terminal kinase; SOCS3, suppressor of cytokine signaling 3; IL-1β, interleukin 1β; IL-6, interleukin 6; TNF-α, tumor necrosis factor α; CR, cytokine receptor; AT, adipose tissue.

**Figure 4 ijms-23-15330-f004:**
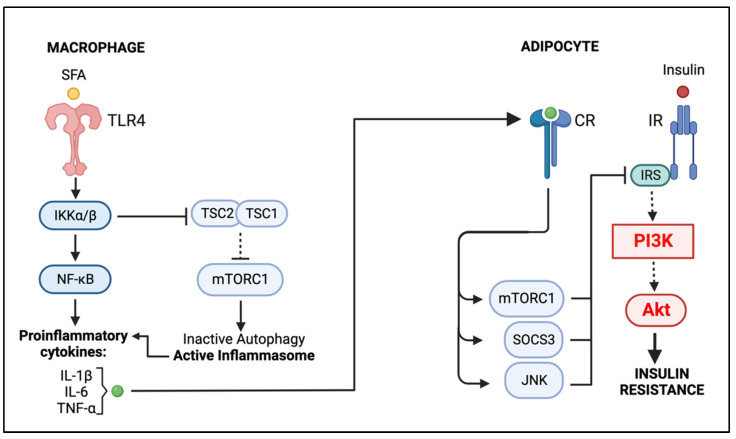
Graphical Summary: Macrophage–Adipocyte Crosstalk in Obese Adipose Tissue. SFAs promote the activation of macrophage NF-κB, which, along with the active inflammasome, induces the secretion of proinflammatory cytokines. These cytokines disrupt adipocyte insulin signaling by inhibiting the IRS1-PI3K/AKT pathway. (Created with BioRender.com on 20 November 2022) Abbreviations: SFAs, saturated fatty acids; TLR4, Toll-like receptor 4; IKK, IκB kinase; IκB, inhibitor κB kinase; NF-κB, Nuclear factor kappa-light-chain-enhancer of activated B cells; TSC1/2, tuberous sclerosis protein complex 1 and 2; mTORC1, mammalian target of rapamycin complex; IR, insulin receptor; IRS, Insulin receptor substrate; PI3K, phosphoinositide 3-kinase; JNK, Jun N-terminal kinase; SOCS3, suppressor of cytokine signaling 3; IL-1β, interleukin 1β; IL-6, interleukin 6; TNF-α, tumor necrosis factor α; CR, cytokine receptor.

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
