# Peer review of "The PI3K/Akt Pathway in Meta-Inflammation"

_ijms, 2022, doi:10.3390/ijms232315330_

Round 1

Reviewer 1 Report

This review is thorough and very well written. However I have trouble with the order of the different sections. I would suggest to bring the general part on insulin and PI3K signalling (section 4) earlier since it is referred to beforehand several times in the present manuscript. This would give an overview of the "healthy state" of the pathway in adipocytes and macrophages before the diseased state is explained.

Here are additional minor comments to consider:

-line 38: what is meant by “sterile inflammation”? This is unclear.

- refer to figure 1 page 8, section 4

- refer to figure 3 page 9, section 4.1

- line 418: figure 2 should be figure 3

Author Response

Thanks.

Reviewer 2 Report

Interesting concept-based review article but it is too lengthy. There are some potential issues and that to be fixed before accepting this article in this journal.

Critical comments:

1. It is not clear from this article, what is the primary aim to write this article?

2. Authors discussed or focused on variety of adipocytes macrophages, I would suggest changing the Title of the article and add "macrophage" in the title

3. Obesity/inflammation/PI3K pathway are all related to cancer and authors never mentioned about cancer in their article

4. Is there any involvement of obesity/inflammation-induced transcriptional activity via JAK-STAT pathway?  Since authors mentioned the of SOCS3 in their article and SOCS family protein is a negative regulatory factor for cytokine/IL-mediated JAK-STST pathway

5. Authors may provide a flow chart of the crosstalk between macrophages and adipocytes. It will be helpful for readers to flow the article

6. Since this article based on metabolic role of PI3K, authors should include reference of Prof. Cantley L.

7. Macrophage section is too big, it should be edited

8. Section 4 1st part (line 326-356) should be delated, there is no new information.

8. Section 5, macrophages-adipocytes crosstalk is way big, prpper editing needed

9. In the conclusion authors may add one or two sentences related to future direction

Author Response

Thanks.
